# The Prognostic Impact of ABO Blood Group in Hepatocellular Carcinoma Following Hepatectomy

**DOI:** 10.3390/cancers15112905

**Published:** 2023-05-25

**Authors:** Masaki Kaibori, Kengo Yoshii, Kosuke Matsui, Hideyuki Matsushima, Hisashi Kosaka, Hidekazu Yamamoto, Takayoshi Nakajima, Kazunori Aoi, Takashi Yamaguchi, Katsunori Yoshida, Mitsugu Sekimoto

**Affiliations:** 1Department of Surgery, Kansai Medical University, Hirakata 573-1010, Japan; matsuik@hirakata.kmu.ac.jp (K.M.);; 2Department of Mathematics and Statistics in Medical Sciences, Kyoto Prefectural University of Medicine, Kyoto 606-0823, Japan; yoshii-k@koto.kpu-m.ac.jp; 3Department of Surgery, Meiwa Hospital, Nishinomiya 663-8186, Japan; 4Department of Gastroenterology and Hepatology, Kansai Medical University, Hirakata 573-1010, Japan

**Keywords:** hepatocellular carcinoma, hepatectomy, Japanese patients, ABO blood group, type A, non-type A, propensity score matching, risk factor, recurrence, long-term outcomes

## Abstract

**Simple Summary:**

The aim of the present study is to determine the prognostic impact of ABO blood types on the survival of a Japanese population of patients with HCC who underwent surgical resection. We retrospectively analyzed 480 patients with HCC who had R0 resection between 2010 and 2020. Outcomes for type A (*n* = 173) and non-type A (*n* = 173) groups after surgery were compared using 1-to-1 propensity score matching to control for variables. Recurrence-free survival (RFS; hazard ratio [HR] 0.75, 95% confidence interval [Cl] 0.58–0.98, *p* = 0.038) and overall survival (OS; HR: 0.67, 95% Cl: 0.48–0.95, *p* = 0.023) for patients with blood type A were both significantly decreased relative to non-type A patients. Cox proportional hazard analysis demonstrated that patients with HCC who have blood type A had a worse prognosis than those with non-type A blood. ABO blood type may have a prognostic impact for patients with HCC after hepatectomy.

**Abstract:**

Background/Purpose: The effect of the ABO blood group on the survival of patients with hepatocellular carcinoma (HCC) is unclear. The aim of the present study is to determine the prognostic impact of ABO blood types on the survival of a Japanese population of patients with HCC who underwent surgical resection. Methods: Patients with HCC (*n* = 480) who underwent an R0 resection between 2010 and 2020 were retrospectively analyzed. Survival outcomes were investigated according to ABO blood type (A, B, O, or AB). Outcomes for type A (*n* = 173) and non-type A (*n* = 173) groups after surgery were compared using 1-to-1 propensity score matching to control for variables. Results: In the study cohort, 173 (36.0%), 133 (27.7%), 131 (27.3%), and 43 (9.0%) of participants had Type A, O, B, and AB, respectively. Type A and non-type A patients were successfully matched based on liver function and tumor characteristics. Recurrence-free survival (RFS; hazard ratio [HR] 0.75, 95% confidence interval [Cl] 0.58–0.98, *p* = 0.038) and overall survival (OS; HR: 0.67, 95% Cl: 0.48–0.95, *p* = 0.023) for patients with blood type A were both significantly decreased relative to non-type A patients. Cox proportional hazard analysis demonstrated that patients with HCC who have blood type A had a worse prognosis than those with non-type A blood. Conclusion: ABO blood type may have a prognostic impact on patients with HCC after hepatectomy. Blood type A is an independent unfavorable prognostic factor for recurrence-free and overall survival (RFS and OS) after hepatectomy.

## 1. Introduction

Hepatocellular carcinoma (HCC) is one of the most common malignancies worldwide, with a high prevalence in Asia and Africa and an increasing prevalence in Western countries [1]. Advances in surgical techniques and perioperative management have transformed HCC resection into a relatively safe operation with a low mortality rate [2]. Liver resection is now accepted as a first-line treatment for HCC in patients with preserved hepatic function [3,4]. However, long-term survival remains unsatisfactory because of the high recurrence rate of HCC after curative hepatectomy [5,6,7]. Effective therapy and disease surveillance for patients with HCC requires prognostic factors used in clinical practice, such as stage and histological grade, and levels of α-fetoprotein (AFP) and protein induced by vitamin K absence or antagonist-II (PIVKA-II) [8,9,10,11]. Recently, findings indicated that ABO blood type is important not only for transfusions but also has clinical implications for various diseases. ABO blood types are reported to be associated with susceptibility to cardiovascular disease, thrombosis, and risk and prognosis of cancer [12,13,14]. For HCC, the impact and prognostic ability of ABO blood type are unclear. One study of prognostic indicators for patients with HCC following hepatectomy indicated that non-type O blood groups were associated with a poorer prognosis, while another found increased HCC risk compared to that for patients with type-O blood [15,16]. In the Korean population, blood group A was associated with an increased risk of developing HCC [17]. The aim of the present study was to determine the prognostic impact of ABO blood types for Japanese patients with HCC who underwent surgical resection.

## 2. Materials and Methods

### 2.1. Patients

The records for all patients with HCC who underwent liver resection between January 2010 and September 2020 at the Kansai Medical University Hospital (Osaka, Japan) were screened. A total of 480 patients with HCC underwent an R0 resection, defined as macroscopic removal of all tumors, of which 429 were classified as Child-Pugh A, and were enrolled. Patients’ characteristics, laboratory and pathological data, treatment details, and ABO blood type (A, B, O, or AB) were retrospectively analyzed from a prospectively collected database. A single surgeon who has performed over 1500 hepatic resections treated all cases referenced in this study. The study protocol was approved by the institutional ethics committee of Kansai Medical University (reference number: KMU 2022211).

### 2.2. Underlying Liver Disease and Liver Function

HCC cases that were positive for anti-HCV and -HBV surface antigen were classified as being due to HCV and HBV, respectively. Patients reporting alcohol abuse (≥60 g/day) with liver disease were classified as having underlying alcohol-related liver disease [18]. Child-Pugh score/classification [19], albumin-bilirubin (ALBI) grade [20], and fibrosis-4 (FIB-4) index [21] were used to assess hepatic reserve function.

### 2.3. Clinicopathologic Variables and Treatment Algorithm for HCC

For all study participants, conventional liver function tests, including measurement of alpha-fetoprotein (AFP) and protein induced by vitamin K absence or antagonist-II (PIVKA-II) levels, were carried out, as was the measurement of indocyanine green retention rate at 15 min (ICG-R15). The updated treatment algorithm for HCC that considers liver function reserve, extrahepatic metastasis, vascular invasion, and the number and size of tumors were used [22]. Hepatectomy decisions were based on liver damage (including ICG-R15 measurement). We summarize the new treatment algorithm as follows. One of three treatment regimens was recommended for patients with HCC who had Child–Pugh class A/B liver function without extrahepatic metastasis or vascular invasion. First, for patients having up to three HCC tumors measuring ≤3 cm, surgical resection or radiofrequency ablation was recommended without priority. For solitary HCC tumors of any size, the recommended first-line therapy was surgical resection. Second, for patients having up to three HCC tumors larger than 3 cm, the recommended first-line therapy was surgical resection, and a second-line therapy was transarterial chemoembolization. Third, cases having HCC with vascular invasion in the absence of extrahepatic metastasis were recommended for combined embolization, hepatectomy, and hepatic arterial infusion chemotherapy, together with molecular targeted therapy. Each patient was treated according to their individual situation with consideration given to liver function, HCC condition, and extent of vascular invasion.

### 2.4. Evaluation of Complications Following Surgical Resection

Complications associated with surgical resection were evaluated based on the Clavien-Dindo classification [23]; significant complications had a grade ≥3.

### 2.5. Propensity Score Analysis

To avoid confounding differences due to baseline varieties between blood type A and non-type A, we established a propensity score-matched subset. Propensity score analysis was used to build matched groups of patients to compare oncological and short- and long-term outcomes between the two groups. Propensity scores were generated using preoperative characteristics, including the American Society of Anesthesiologists-physical status (ASA-PS), AFP, and ALBI scores. Propensity scores were matched using a caliper width of 0.6 multiplied by the standard deviation of values calculated by a logistic regression model. Each patient in the blood type A group was matched to one patient in the non-type A group using a greedy nearest-neighbor matching algorithm without replacement.

### 2.6. Statistical Analysis

Continuous variables were classified into two categories using the median value for the samples. The chi-square test or Fisher’s exact test were used to compare four or two groups of clinical characteristics as appropriate. The Kaplan–Meier method was used to calculate the RFS and OS probability after hepatectomy. Univariate Cox analysis was used to estimate RFS and OS hazard ratios and 95% confidence intervals (Cis). Multivariate analysis was carried out using Cox proportional hazards analysis. Statistical significance was indicated by *p* values ≤ 0.05 for all analyses. R version 4.1.2 (R Foundation for Statistical Computing, Vienna, Austria) was used for all statistical analyses, and the “survival” package within R was used for survival analysis.

## 3. Results

### 3.1. Patient Selection and Characteristics

Characteristics of the 480 patients who underwent hepatic resection are presented in Table 1. Of the 480 participants, 173 (36.0%), 133 (27.7%), 131 (27.3%), and 43 (9.0%) had ABO blood type A, O, B, and AB, respectively. The blood groups had similar baseline characteristics, except that patients with blood types A and AB had higher AFP levels (*p* = 0.029) and were more likely to have multiple tumors (*p* = 0.026).

We classified patients into two groups, type A and non-type A. Table 2 summarizes the perioperative characteristics of both groups before and after propensity score matching (PSM). Since the two groups had significantly different preoperative AFP levels before PSM, we classified patients into type A and non-type A groups (*n* = 173 each; Table 2) using paired PSM (*n* = 173 each; Table 2). After matching, there was no significant difference between the two groups in terms of preoperative background variables, surgical factors, or clinicopathologic features.

### 3.2. Long-Term Survival

Type B patients had the highest 5-year RFS rate (37.6%), followed by type O (28.9%) and type AB (23.4%). Type A patients had the lowest RFS (21.7%; Figure 1A). The hazard ratios with type A for RFS were 0.75 for type O (95% CI: 0.56–1.00; *p* = 0.048), 0.66 for type B (95% CI: 0.49–0.89; *p* = 0.007), and 0.83 for type AB (95% CI: 0.56–1.24; *p* = 0.365) (Figure 1A). Type A also had the lowest 5-year OS rate (46.3%), while type O had the highest (64.2%), followed by type AB (61.3%) and B (60.4%; Figure 1B). The hazard ratios with type A for OS were 0.61 [95% CI: 0.42–0.89; *p* = 0.011], 0.68 [95% CI: 0.47–0.99; *p* = 0.043], and 0.79 [95% CI: 0.47–1.32; *p* = 0.372] for O, B, and AB, respectively (Figure 1B).

The type A and non-type A groups had 5-year RFS rates of 31.5% and 21.7%, respectively (Figure 2A), and the 5-year OS rates were 62.1% and 46.3%, respectively (Figure 2B). These differences were significant (RFS: HR, 0.73; 95% CI, 0.58–0.92; *p* = 0.007; Figure 2A and OS: HR, 0.67 (95% CI, 0.50–0.90; *p* = 0.007) (Figure 2B). For PS-adjusted long-term outcomes, the RFS and OS also differed significantly between the type A and non-type A groups (Figure 2). The 5-year RFS rates were 21.7% for type A and 29.7% for non-type A (HR: 0.75, 95% CI, 0.58–0.98, *p* = 0.038; Figure 2C). The 5-year OS rates were 46.3% for type A and 60.9% for non-type A (HR: 0.67, 95% CI, 0.48–0.95, *p* = 0.023; Figure 2D).

We also performed a survival analysis for O type and non-O type patients before and after PSM. The 5-year RFS rates for type O and non-type O groups were 28.9% and 27.5%, respectively (Appendix A), and the 5-year OS rates were 64.2% and 53.6%, respectively (Appendix A). These differences were not significant (RFS: HR 1.12, 95% CI 0.87–1.45, *p* = 0.386, Appendix A, OS: HR 1.39, 95% CI 0.98–1.97, *p* = 0.065, Appendix A). Analyses using PS-adjusted data also showed no significant difference in RFS and OS between the type O and non-type O groups (Appendix A). The 5-year RFS rate was 28.3% for type O, and 24.9% for non-type O (HR: 1.15, 95% CI: 0.83–1.61, *p* = 0.405, Appendix A), and the 5-year OS rate was 60.3% for type O type and 51.8% for non-type O (HR: 1.38, 95% CI 0.88–2.16, *p* = 0.158, Appendix A).

We performed several subgroup analyses of RFS and OS between the type A and non-type A groups (Figure 3). Among patients younger than 73 years old or patients with ASA-PS class II disease, Forest plots showed that non-type A patients who underwent hepatic resection had better RFS and OS than those who were type A. Among patients with serum C-reactive protein (CRP) < 0.1 mg/dL, Child-Pugh ≤ 6, tumor size ≤ 3.5 cm, solitary tumor, or negative portal vein invasion, RFS and OS were significantly better in the non-type A group than the type A group.

### 3.3. Examination of Prognostic Factors for Long-Term Survival by Univariate and Multivariate Analyses

Cox proportional hazards analysis of RFS and OS indicated that there were five independent prognostic predictors (Table 3): PIVKA-II ≥ 109 mAU/mL (RFS: hazard ratio, 1.59; 95% CI, 1.15–2,19; *p* = 0.005, OS: hazard ratio, 1.71; 95% CI, 1.11–2.63; *p* = 0.014), positive portal vein invasion (RFS: hazard ratio, 1.55; 95% CI, 1.10–2.17; *p* = 0.011, OS: hazard ratio, 1.79; 95% CI, 1.11–2.87; *p* = 0.016), and non-type A blood type (RFS: hazard ratio, 0.73; 95% CI, 0.54–0.98; *p* = 0.035, OS: hazard ratio, 0.58; 95% CI, 0.39–0.86; *p* = 0.006).

## 4. Discussion

The present study is the first to show the influence of ABO blood type in a large cohort of Japanese patients with HCC following hepatectomy. The type A blood group was predicted to have poorer RFS and OS compared to non-type A blood groups (Figure 2; Table 3). Among younger patients and patients with good PS, few inflammatory reactions or good liver function, and unremarkable oncological behavior, RFS and OS were significantly better for the non-type A group than the type A group (Figure 3).

The role of inherited blood group antigens in cancer risk and progression has been examined for many types of solid organ tumors [24,25,26,27,28,29,30,31,32], but few studies assessed how the ABO blood group impacts the survival of patients with HCC. We are currently aware of only one investigation that considered how the ABO blood group is related to the prognosis for patients with HCC following hepatectomy [15]. Wu et al. demonstrated that patients with non-O blood type (i.e., Type A, B, and AB) had reduced OS compared to Type O patients with HCC in China, which is in contrast to our results. This difference could be due in part to differences in the blood type proportions in that our study population had 7% more and 10% fewer patients with Type A and O, respectively (36.0% vs. 28.8% and 27.7% and 37.9%). Shin et al. also reported an association with blood type in the first diagnosis of HCC among patients treated at a single hospital in South Korea [17]. In a case–control study of 1538 patients with newly diagnosed HCC at their hospital and 1305 randomly selected members of the general population, blood type A and genotype A had the highest risks for HCC. No significant difference was seen among AO, BO, BB, and AB genotypes or blood groups B and AB.

Results of earlier studies indicated that blood type A is associated with an increased risk of various cancers, including stomach [27], ovarian [28], and pancreatic [29]. Meanwhile, the risk of renal cell [30], colorectal [31], and skin [32] cancer was increased for non-type O blood groups. The mechanisms responsible for these differences in risk are currently unclear. Previous studies demonstrated that the proliferation and motility of colon tumor cells are highly associated with the expression of ABO blood type antigen, particularly blood type antigen A [33,34]. Moreover, Marionneau et al. showed in rat colon carcinoma cells that the expression of A antigen increases resistance to apoptosis and facilitates escape from immune control [35]. These data suggest a direct involvement of ABO blood type antigens in the development and metastasis of colorectal cancer. Several findings suggested that the structure of certain tumor antigens is similar to that of ABO antigens. Smith and Prieto [36] showed that the Forssmann antigen, which is predominantly present in stomach and colon tumors, is almost structurally identical to the A antigen determinant. Okada et al. found neoexpression of the ABH blood group antigens in HCC tissues [37]. Taken together, these results suggest that patients with blood group A may have diminished tumor immune response due to the reduced ability of the immune system to recognize and attack tumor cells expressing antigens that are structurally similar to the ABO antigen [38]. A limited number of earlier studies to examine how Type ABO impacts HCC produced differing results. Results of our study indicated that patients with HCC who had type A blood had poorer prognoses than patients who had non-type A blood. First, blood antigens perform important roles as receptors or ligands for microbes and immunologically important proteins [39,40] that are integral to the malignant progression and spread of cancer [41]. Abnormal expression of ABO blood antigens in liver tissue may be related to HCC carcinogenesis. The ABO blood antigens (A, B, and H) are typically expressed on the surface of red blood cells (RBCs) and most epithelial tissues but not on hepatocytes in a normal liver. A previous study indicated increased ABH expression or neo-expression in HCC tissues [42], suggesting that ABO antigens or ABO antigen expression might perform a role in HCC carcinogenesis. Another study reported that non-type O blood is an independent risk factor for liver fibrosis progression related to HCV infection [43]. Patients with blood type A tended to have greater impairment of liver function and earlier cirrhosis onset compared to those with blood type O [44].

## 5. Limitations

The present study has some limitations. This was a retrospective, single-institution study with a limited number of patients, almost all of whom were Japanese, which may have caused a selection bias that affected the results. Our evaluation of phenotype, but not the ABO blood type genotype or ABO allele subtype, may also have affected the results. Further investigations of different racial groups in multicenter studies across multiple countries are needed to generalize our findings.

## 6. Conclusions

In conclusion, the results of this study show an association between ABO blood type and the prognosis of Japanese patients with HCC after hepatectomy. In particular, blood type A is an independent unfavorable prognostic factor for RFS and OS in patients with HCC that undergo hepatectomy.

## Figures and Tables

**Figure 1 cancers-15-02905-f001:**
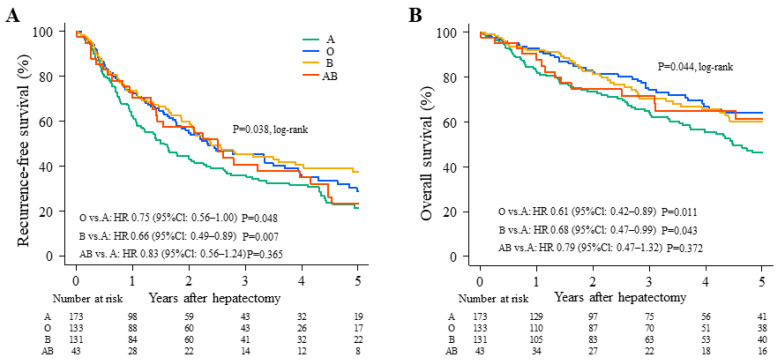
Comparison of recurrence-free survival (RFS) and overall survival (OS) among four ABO blood types. (**A**) RFS and (**B**) OS according to ABO blood type (A: Green; O: Blue; B: Yellow; AB: Red). CI, confidence interval; HR, hazard ratio.

**Figure 2 cancers-15-02905-f002:**
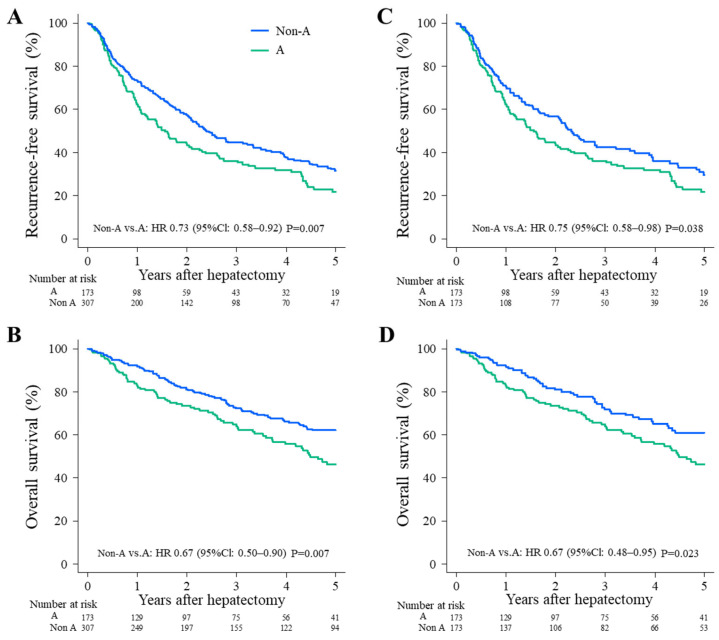
Survival outcomes. Comparison of survival outcomes after hepatic resection for type A (green) and non-type A patients (blue). (**A**) RFS and (**B**) OS before propensity score matching. (**C**) RFS and (**D**) OS after propensity score matching. HR, hazard ratio; CI, confidence interval.

**Figure 3 cancers-15-02905-f003:**
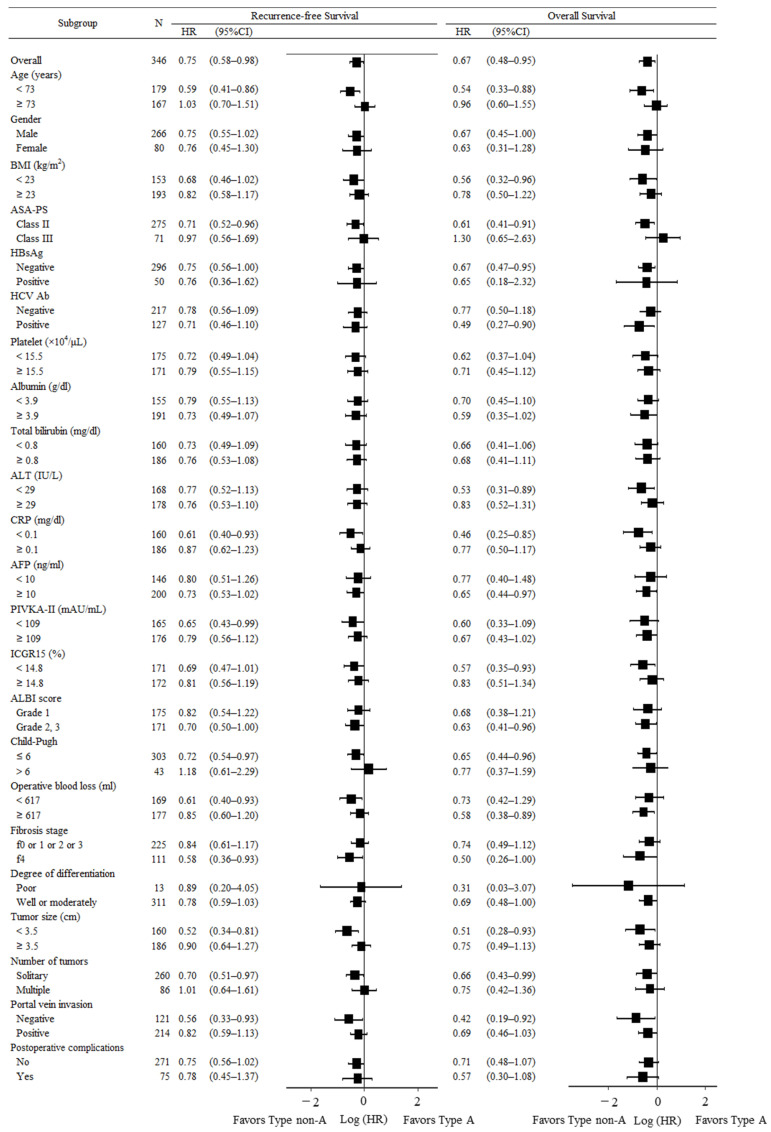
Recurrence-free survival and overall survival in selected subgroups. Hazard ratios were compared between the type A and non-type A groups in selected subgroups. CI, confidence interval; HR, hazard ratio; BMI: body mass index; ASA-PS: American Society of Anesthesiologists-physical status; HBsAg: hepatitis B surface antigen; HCVAb: hepatitis C virus antibody; ALT: alanine aminotransferase; CRP: C-reactive protein; AFP: α-fetoprotein; PIVKA-II: protein induced by vitamin K absence or antagonist-II; ICGR15: indocyanine green retention rate at 15 min; ALBI: integration of albumin-bilirubin.

**Table 1 cancers-15-02905-t001:** Clinicopathological characteristics stratified by ABO blood type.

Variable	Blood Type A (*n* = 173)		Blood Type O (*n* = 133)		Blood Type B (*n* = 131)		Blood Type AB (*n* = 43)		*p*
Age (years)									0.126
<73	81	(47%)	67	(50%)	72	(55%)	15	(35%)	
≥73	92	(53%)	66	(50%)	59	(45%)	28	(65%)	
Gender									0.096
Male	135	(78%)	105	(79%)	89	(68%)	35	(81%)	
Female	38	(22%)	28	(21%)	42	(32%)	8	(19%)	
BMI (kg/m^2^)									0.109
<23	73	(42%)	54	(41%)	71	(54%)	20	(47%)	
≥23	100	(58%)	79	(59%)	60	(46%)	23	(53%)	
ASA-PS									0.407
Class I	0	(0%)	3	(2%)	4	(3%)	1	(2%)	
Class II	138	(80%)	106	(80%)	105	(81%)	32	(74%)	
Class III	35	(20%)	24	(18%)	21	(16%)	10	(23%)	
Alcohol									0.156
None	118	(68%)	80	(60%)	95	(73%)	31	(72%)	
Positive	55	(32%)	53	(40%)	36	(27%)	12	(28%)	
Diabetes mellitus									0.790
None	115	(66%)	90	(68%)	90	(69%)	26	(60%)	
Positive	58	(34%)	43	(32%)	41	(31%)	17	(40%)	
Hypertension									0.825
None	90	(52%)	71	(53%)	75	(57%)	24	(56%)	
Positive	83	(48%)	62	(47%)	56	(43%)	19	(44%)	
HBsAg									0.598
Negative	148	(86%)	116	(87%)	112	(85%)	40	(93%)	
Positive	25	(14%)	17	(13%)	19	(15%)	3	(7%)	
HCV Ab									0.807
Negative	110	(64%)	82	(62%)	81	(62%)	24	(56%)	
Positive	62	(36%)	50	(38%)	50	(38%)	19	(44%)	
Platelet (×10^4^/μL)									0.732
<15.5	88	(51%)	65	(49%)	61	(47%)	24	(56%)	
≥15.5	85	(49%)	68	(51%)	70	(53%)	19	(44%)	
Albumin (g/dL)									0.354
<3.9	79	(46%)	50	(38%)	56	(43%)	22	(51%)	
≥3.9	94	(54%)	83	(62%)	75	(57%)	21	(49%)	
Total bilirubin (mg/dL)									0.470
<0.8	84	(49%)	59	(44%)	70	(53%)	23	(53%)	
≥0.8	89	(51%)	74	(56%)	61	(47%)	20	(47%)	
ALT (IU/L)									0.849
<29	88	(51%)	63	(47%)	65	(50%)	19	(44%)	
≥29	85	(49%)	70	(53%)	66	(50%)	24	(56%)	
Prothrombin time (%)									0.180
<87	86	(50%)	58	(44%)	64	(49%)	27	(63%)	
≥87	86	(50%)	75	(56%)	67	(51%)	16	(37%)	
CRP (mg/dL)									0.946
<0.1	82	(47%)	67	(50%)	65	(50%)	22	(51%)	
≥0.1	91	(53%)	66	(50%)	66	(50%)	21	(49%)	
Cholinesterase (U/L)									0.661
<225	92	(53%)	61	(46%)	63	(48%)	22	(51%)	
≥225	81	(47%)	71	(54%)	67	(52%)	21	(49%)	
AFP (ng/mL)									0.029
<10	74	(43%)	73	(55%)	77	(59%)	20	(47%)	
≥10	99	(57%)	60	(45%)	54	(41%)	23	(53%)	
PIVKA-II (mAU/mL)									0.156
<109	84	(49%)	73	(58%)	56	(44%)	21	(50%)	
≥109	87	(51%)	53	(42%)	72	(56%)	21	(50%)	
ICGR15 (%)									0.524
<14.8	80	(47%)	72	(55%)	64	(49%)	20	(47%)	
≥14.8	91	(53%)	59	(45%)	67	(51%)	23	(53%)	
ALBI score									0.308
Grade 1	86	(50%)	76	(57%)	72	(55%)	18	(42%)	
Grade 2	81	(47%)	55	(41%)	58	(44%)	25	(58%)	
Grade 3	6	(3%)	2	(2%)	1	(1%)	0	(0%)	
FIB4-index									0.471
Low	6	(3%)	9	(7%)	11	(8%)	4	(9%)	
Middle	60	(35%)	44	(33%)	45	(34%)	11	(26%)	
High	106	(62%)	80	(60%)	75	(57%)	28	(65%)	
Child-Pugh score									0.056
≤6	150	(87%)	120	(90%)	122	(93%)	34	(79%)	
>6	23	(13%)	13	(10%)	9	(7%)	9	(21%)	
Esophageal/gastric varices									0.064
Negative	136	(89%)	94	(79%)	107	(87%)	36	(92%)	
Positive	17	(11%)	25	(21%)	16	(13%)	3	(8%)	
Operative blood loss (mL)									0.073
<617	90	(52%)	73	(55%)	63	(48%)	14	(33%)	
≥617	83	(48%)	60	(45%)	68	(52%)	29	(67%)	
Operative time (min)									0.233
<331	95	(55%)	69	(52%)	59	(45%)	18	(42%)	
≥331	78	(45%)	64	(48%)	72	(55%)	25	(58%)	
Fibrosis stage									0.570
f0 or 1 or 2 or 3	115	(68%)	92	(71%)	81	(63%)	30	(70%)	
f4	54	(32%)	37	(29%)	47	(37%)	13	(30%)	
Degree of differentiation									0.940
Poor	7	(4%)	4	(3%)	6	(5%)	1	(3%)	
Well or moderately	154	(96%)	118	(97%)	119	(95%)	38	(97%)	
Tumor size (cm)									0.499
<3.5	85	(49%)	63	(47%)	55	(42%)	23	(53%)	
≥3.5	88	(51%)	70	(53%)	76	(58%)	20	(47%)	
Number of tumors									0.026
Solitary	128	(74%)	107	(80%)	112	(85%)	29	(67%)	
Multiple	45	(26%)	26	(20%)	19	(15%)	14	(33%)	
Portal vein invasion									0.639
Negative	68	(40%)	49	(39%)	49	(38%)	12	(29%)	
Positive	101	(60%)	78	(61%)	80	(62%)	29	(71%)	
Hepatic vein invasion									0.915
Negative	111	(66%)	84	(67%)	82	(65%)	25	(61%)	
Positive	57	(34%)	42	(33%)	45	(35%)	16	(39%)	
Readmission within 30 days									0.884
No	159	(94%)	120	(92%)	122	(94%)	39	(95%)	
Yes	11	(6%)	11	(8%)	8	(6%)	2	(5%)	
Postoperative complications (Clavien-Dindo classification ≥ 3)									0.680
No	136	(79%)	105	(79%)	106	(81%)	31	(72%)	
Yes	37	(21%)	28	(21%)	25	(19%)	12	(28%)	

Data are shown as *n* (%). BMI: body mass index; ASA-PS: American Society of Anesthesiologists-physical status; HBsAg: hepatitis B surface antigen; HCVAb: hepatitis C virus antibody; ALT: alanine aminotransferase; CRP: C-reactive protein; AFP: α-fetoprotein; PIVKA-II: protein induced by vitamin K absence or antagonist-II; ICGR15: indocyanine green retention rate at 15 min; ALBI: integration of albumin-bilirubin; FIB-4 index: fibrosis-4 index.

**Table 2 cancers-15-02905-t002:** Clinicopathological characteristics according to blood type A vs. non-type A before and after propensity score matching (PSM).

	Before PSM (*n* = 480)					After PSM (*n* = 346)				
Variables	Blood Type A (*n* = 173)		Non-A Blood Type (*n* = 307)		*p*	Blood Type A (*n* = 173)		Non-A Blood Type (*n* = 173)		*p*
Age (years)					0.543					0.085
<73	81	(47%)	154	(50%)		81	(47%)	98	(57%)	
≥73	92	(53%)	153	(50%)		92	(53%)	75	(43%)	
Gender					0.463					0.702
Male	135	(78%)	229	(75%)		135	(78%)	131	(76%)	
Female	38	(22%)	78	(25%)		38	(22%)	42	(24%)	
BMI (kg/m^2^)					0.333					0.516
<23	73	(42%)	145	(47%)		73	(42%)	80	(46%)	
≥23	100	(58%)	162	(53%)		100	(58%)	93	(54%)	
ASA-PS					0.081					1.000
Class I	0	(0%)	8	(3%)		0	(0%)	0	(0%)	
Class II	138	(80%)	243	(79%)		138	(80%)	137	(79%)	
Class III	35	(20%)	55	(18%)		35	(20%)	36	(21%)	
Alcohol use					0.883					0.569
None	118	(68%)	206	(67%)		118	(68%)	112	(65%)	
Positive	55	(32%)	101	(33%)		55	(32%)	61	(35%)	
Diabetes mellitus					0.969					1.000
None	115	(66%)	206	(67%)		115	(66%)	116	(67%)	
Positive	58	(34%)	101	(33%)		58	(34%)	57	(33%)	
Hypertension					0.540					0.159
None	90	(52%)	170	(55%)		90	(52%)	104	(60%)	
Positive	83	(48%)	137	(45%)		83	(48%)	69	(40%)	
HBsAg					0.689					1.000
Negative	148	(86%)	268	(87%)		148	(86%)	148	(86%)	
Positive	25	(14%)	39	(13%)		25	(14%)	25	(14%)	
HCV Ab					0.605					0.823
Negative	110	(64%)	187	(61%)		110	(64%)	107	(62%)	
Positive	62	(36%)	119	(39%)		62	(36%)	65	(38%)	
Platelet (×10^4^/μL)					0.744					1.000
<15.5	88	(51%)	150	(49%)		88	(51%)	87	(50%)	
≥15.5	85	(49%)	157	(51%)		85	(49%)	86	(50%)	
Albumin (g/dL)					0.455					0.829
<3.9	79	(46%)	128	(42%)		79	(46%)	76	(44%)	
≥3.9	94	(54%)	179	(58%)		94	(54%)	97	(56%)	
Total bilirubin (mg/dL)					0.915					0.450
<0.8	84	(49%)	152	(50%)		84	(49%)	76	(44%)	
≥0.8	89	(51%)	155	(50%)		89	(51%)	97	(56%)	
ALT (IU/L)					0.594					0.452
<29	88	(51%)	147	(48%)		88	(51%)	80	(46%)	
≥29	85	(49%)	160	(52%)		85	(49%)	93	(54%)	
Prothrombin time (%)					0.832					0.555
<87	86	(50%)	149	(49%)		86	(50%)	93	(54%)	
≥87	86	(50%)	158	(51%)		86	(50%)	80	(46%)	
CRP (mg/dL)					0.627					0.746
<0.1	82	(47%)	154	(50%)		82	(47%)	78	(45%)	
≥0.1	91	(53%)	153	(50%)		91	(53%)	95	(55%)	
Cholinesterase (U/L)					0.307					0.789
<225	92	(53%)	146	(48%)		92	(53%)	88	(51%)	
≥225	81	(47%)	159	(52%)		81	(47%)	84	(49%)	
AFP (ng/mL)					0.011					0.913
<10	74	(43%)	170	(55%)		74	(43%)	72	(42%)	
≥10	99	(57%)	137	(45%)		99	(57%)	101	(58%)	
PIVKA-II (mAU/mL)					0.820					0.870
<109	84	(49%)	150	(51%)		84	(49%)	81	(48%)	
≥109	87	(51%)	146	(49%)		87	(51%)	89	(52%)	
ICGR15 (%)					0.413					0.305
<14.8	80	(47%)	156	(51%)		80	(47%)	91	(53%)	
≥14.8	91	(53%)	149	(49%)		91	(53%)	81	(47%)	
ALBI score					0.126					0.637
Grade 1	86	(50%)	166	(54%)		86	(50%)	89	(51%)	
Grade 2	81	(47%)	138	(45%)		81	(47%)	81	(47%)	
Grade 3	6	(3%)	3	(1%)		6	(3%)	3	(2%)	
FIB4-index					0.170					0.388
Low	6	(3%)	24	(8%)		6	(3%)	10	(6%)	
Middle	60	(35%)	100	(33%)		60	(35%)	51	(29%)	
High	106	(62%)	183	(60%)		106	(62%)	112	(65%)	
Child-Pugh score					0.361					0.745
≤6	150	(87%)	276	(90%)		150	(87%)	153	(88%)	
>6	23	(13%)	31	(10%)		23	(13%)	20	(12%)	
Esophageal/gastric varices					0.247					0.248
Negative	136	(89%)	237	(84%)		136	(89%)	134	(84%)	
Positive	17	(11%)	44	(16%)		17	(11%)	26	(16%)	
Operative blood loss (mL)					0.568					0.282
<617	90	(52%)	150	(49%)		90	(52%)	79	(46%)	
≥617	83	(48%)	157	(51%)		83	(48%)	94	(54%)	
Fibrosis stage					1.000					0.758
f0 or 1 or 2 or 3	115	(68%)	203	(68%)		115	(68%)	110	(66%)	
f4	54	(32%)	97	(32%)		54	(32%)	57	(34%)	
Degree of differentiation					0.993					0.982
Poor	7	(4%)	11	(4%)		7	(4%)	6	(4%)	
Well or moderately	154	(96%)	275	(96%)		154	(96%)	157	(96%)	
Tumor size (cm)					0.562					0.332
<3.5	85	(49%)	141	(46%)		85	(49%)	75	(43%)	
≥3.5	88	(51%)	166	(54%)		88	(51%)	98	(57%)	
Number of tumors					0.105					0.709
Solitary	128	(74%)	248	(81%)		128	(74%)	132	(76%)	
Multiple	45	(26%)	59	(19%)		45	(26%)	41	(24%)	
Portal vein invasion					0.559					0.142
Negative	68	(40%)	110	(37%)		68	(40%)	53	(32%)	
Positive	101	(60%)	187	(63%)		101	(60%)	113	(68%)	
Hepatic vein invasion					0.890					0.642
Negative	111	(66%)	191	(65%)		111	(66%)	104	(63%)	
Positive	57	(34%)	103	(35%)		57	(34%)	61	(37%)	
Readmission within 30 days					0.992					1.000
No	159	(94%)	281	(93%)		159	(94%)	158	(93%)	
Yes	11	(6%)	21	(7%)		11	(6%)	12	(7%)	
Postoperative complications (Clavien-Dindo classification ≥ 3)					1.000					1.000
No	136	(79%)	242	(79%)		136	(79%)	135	(78%)	
Yes	37	(21%)	65	(21%)		37	(21%)	38	(22%)	

Data are shown as *n* (%). PSM: propensity score matching; BMI: body mass index; ASA-PS: American Society of Anesthesiologists-physical status; HBsAg: hepatitis B surface antigen; HCVAb: hepatitis C virus antibody; ALT: alanine aminotransferase; CRP: C-reactive protein; AFP: α-fetoprotein; PIVKA-II: protein induced by vitamin K absence or antagonist-II; ICGR15: indocyanine green retention rate at 15 min; ALBI: integration of albumin-bilirubin; FIB-4 index: fibrosis-4 index.

**Table 3 cancers-15-02905-t003:** Recurrence-free and overall survival for patients with hepatocellular carcinoma who received hepatic resection as analyzed by Cox proportional hazards regression.

Variable	Recurrence-Free Survival			Overall Survival		
	HR	(95% CI)	*p*	HR	(95% CI)	*p*
Age (≥ vs. <73 years)	1.05	(0.76–1.43)	0.781	1.14	(0.76–1.70)	0.522
BMI (≥ vs. <23 kg/m^2^)	1.07	(0.78–1.46)	0.687	1.09	(0.72–1.65)	0.692
ASA-PS (Class III vs. II)	1.27	(0.84–1.91)	0.259	1.56	(0.96–2.53)	0.074
HBsAg (Positive vs. Negative)	0.71	(0.44–1.16)	0.171	0.40	(0.19–0.86)	0.019
HCV Ab (Positive vs. Negative)	0.98	(0.70–1.37)	0.887	0.78	(0.50–1.22)	0.280
Platelet (≥ vs. <15.5 × 10^4^/μL)	1.13	(0.82–1.55)	0.472	1.39	(0.90–2.14)	0.136
ALT (≥ vs. <29 IU/L)	0.91	(0.67–1.23)	0.537	0.96	(0.65–1.42)	0.849
Prothrombin time (≥ vs. <87%)	0.76	(0.54–1.07)	0.114	0.73	(0.47–1.14)	0.169
AFP (≥ vs. <10 ng/mL)	1.21	(0.88–1.67)	0.246	1.72	(1.09–2.73)	0.020
PIVKA-II (≥ vs. <109 mAU/mL)	1.59	(1.15–2.19)	0.005	1.71	(1.11–2.63)	0.014
ICGR15 (≥ vs. <14.8%)	0.98	(0.71–1.37)	0.921	1.10	(0.71–1.70)	0.681
ALBI score (Grade 2 vs. 1)	1.47	(1.04–2.07)	0.027	1.57	(0.98–2.50)	0.059
ALBI score (Grade 3 vs. 1)	0.74	(0.20–2.76)	0.653	1.09	(0.27–4.47)	0.905
Child-Pugh score (≥ vs. <6)	1.37	(0.79–2.38)	0.265	1.50	(0.84–2.68)	0.172
Operative blood loss (≥ vs. <617 mL)	1.25	(0.91–1.72)	0.162	1.31	(0.86–2.00)	0.208
Fibrosis stage (f0 or 1 or 2 or 3 vs. f4)	0.98	(0.67–1.43)	0.908	0.78	(0.47–1.29)	0.333
Tumor size (≥ vs. <3.5 cm)	1.08	(0.77–1.51)	0.652	0.95	(0.62–1.48)	0.832
Number of tumors (Multiple vs. Solitary)	1.55	(1.11–2.16)	0.010	1.30	(0.85–1.99)	0.220
Portal vein invasion (Positive vs. Negative)	1.55	(1.10–2.17)	0.011	1.79	(1.11–2.87)	0.016
Postoperative complications (Yes vs. No)	1.26	(0.86–1.84)	0.237	2.11	(1.31–3.39)	0.002
Blood type (Non-A vs. A)	0.73	(0.54–0.98)	0.035	0.58	(0.39–0.86)	0.006

HR; hazard ratio; CI; confidence interval; BMI: body mass index; ASA-PS: American Society of Anesthesiologists-physical status; HBsAg: hepatitis B surface antigen; HCVAb: hepatitis C virus antibody; ALT: alanine aminotransferase; AFP: α-fetoprotein; PIVKA-II: protein induced by vitamin K absence or antagonist-II; ICGR15: indocyanine green retention rate at 15 min; ALBI: integration of albumin-bilirubin.

## Data Availability

Due to the nature of this research, participants in this study could not be contacted about whether the findings could be shared publicly. Thus, supporting data are not available. The datasets generated and analyzed during the current study are not publicly available due to the nature of the research but are available from the corresponding author on reasonable request.

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
