# Peer review of "The Prognostic Impact of ABO Blood Group in Hepatocellular Carcinoma Following Hepatectomy"

_cancers, 2023, doi:10.3390/cancers15112905_

Round 1
Reviewer 1 Report
Kaibori et al. studied the prognosis of patients undergoing hepatectomy for HCC and found the relationship between the blood type A vs non-A and the prognosis of patients. They found that surgical patients with blood type A had worse RFS and OS than patients with non-type A.
1. As the author referred, Wu et al reported patients with blood type O had worse OS than in patients with non-O type. The present results and others show that type A was worse. These results imply that type B or type AB may be associated with better survivals than type A or O, which can theoretically lead to the hypothesis that antigen for A may have some anti-cancer effect. Please research the basic difference of blood types and discuss the anti-cancer effect of antigen for A.
2. Did the author perform the comparison between type O and non-type O? If not, please do it. If did, please mention the results. It may be premature that the authors conclude type A is worse than other types.
can be improved by native speakers
Author Response
Responses to the comments of Reviewer #1
Thank you for your valuable comments.
Major comments
- As the author referred, Wu et al reported patients with blood type O had worse OS than in patients with non-O type. The present results and others show that type A was worse. These results imply that type B or type AB may be associated with better survivals than type A or O, which can theoretically lead to the hypothesis that antigen for A may have some anti-cancer effect. Please research the basic difference of blood types and discuss the anti-cancer effect of antigen for A.
Response
Thank you for raising this important point. We have updated the Discussion to reference several studies that examined the mechanisms associated with a poorer prognosis for patients with cancer who have blood type A.
Discussion (lines 251–263)
Previous studies demonstrated that proliferation and motility of colon tumor cells are highly associated with the expression of ABO blood type antigen, particularly blood type antigen A [33, 34]. Moreover, Marionneau et al. showed in rat colon carcinoma cells that expression of A antigen increases resistance to apoptosis and facilitates escape from immune control [35]. These data suggest a direct involvement of ABO blood type antigens in the development and metastasis of colorectal cancer. Several findings suggested that the structure of certain tumor antigens is similar to that of ABO antigens. Smith and Prieto [36] showed that the Forssmann antigen, which is predominantly present in stomach and colon tumors, is almost structurally identical to the A antigen determinant. Okada et al. found neoexpression of ABH blood group antigens in HCC tissues [37]. Taken together, these results suggest that patients with blood group A may have diminished tumor immune response due to the reduced ability of the immune system to recognize and attack tumor cells expressing antigens that are structurally similar to the ABO antigen [38].
New references
- Labarrière N, Piau JP, Otry C, Denis M, Lustenberger P, Meflah K, Le Pendu J. H blood group antigen carried by CD44V modulates tumorigenicity of rat colon carcinoma cells. Cancer Res. 1994; 54: 6275-81.
- Iwamoto S, Withers DA, Handa K, Hakomori S. Deletion of A-antigen in a human cancer cell line is associated with reduced promoter activity of CBF/NF-Y binding region, and possibly with enhanced DNA methylation of A transferase promoter. Glycoconj J. 1999; 16: 659-66. doi: 10.1023/a:1007085202379.
- Marionneau S, Le Moullac-Vaidye B, Le Pendu J. Expression of histo-blood group A antigen increases resistance to apoptosis and facilitates escape from immune control of rat colon carcinoma cells. Glycobiology. 2002; 12: 851-6. doi: 10.1093/glycob/cwf103.
- Smith DF, Prieto PA. Forssmann antigen. In: Roitt IM, Delves PH (eds). Encyclopedia of Immunology. Academic Press, London 1992; 591-2.
- Okada Y, Arima T, Togawa K, Nagashima H, Jinno K, Moriwaki S, Kunitomo T, Thurin J, Koprowski H. Neoexpression of ABH and Lewis blood group antigens in human hepatocellular carcinomas. J Natl Cancer Inst. 1987; 78: 19-28. doi: 10.1093/jnci/78.1.19.
- Yuzhalin AE, Kutikhin AG. ABO and Rh blood groups in relation to ovarian, endometrial and cervical cancer risk among the population of South-East Siberia. Asian Pac J Cancer Prev. 2012; 13: 5091-6. doi: 10.7314/apjcp.2012.13.10.5091.
- Did the author perform the comparison between type O and non-type O? If not, please do it. If did, please mention the results. It may be premature that the authors conclude type A is worse than other types.
Response
According to the reviewer's suggestion, we performed a survival analyses of O type and non-O type groups before and after PSM. The 5-year RFS rates for type O and non-type O groups were 28.9% and 27.5%, respectively (Supplementary Fig. 1A), and the 5-year OS rates were 64.2% and 53.6%, respectively (Supplementary Fig. 1B). These differences were not significant (RFS: HR 1.12, 95% CI 0.87-1.45, P=0.386, Supple. Fig.1A, OS: HR 1.39, 95% CI 0.98-1.97, P=0.065, Supple. Fig. 1B). Analyses using PS-adjusted data also showed no significant difference in RFS and OS between the type O and non-type O groups (Supplementary Fig. 1C). The 5-year RFS rate was 28.3% for type O and 24.9% for non-type O (HR: 1.15, 95% CI: 0.83-1.61, P = 0.405, Supple. Fig. 1C), and the 5-year OS rate was 60.3% for type O type and 51.8% for non-type O (HR: 1.38, 95% CI 0.88-2.16, P = 0.158, Supple. Fig. 1D). We have made the following update to the Results section.
Results (lines 184–192)
We also performed a survival analysis for O type and non-O type patients before and after PSM. The 5-year RFS rates for type O and non-type O groups were 28.9% and 27.5%, respectively (Supplementary Fig. 1A), and the 5-year OS rates were 64.2% and 53.6%, respectively (Supplementary Fig. 1B). These differences were not significant (RFS: HR 1.12, 95% CI 0.87-1.45, P=0.386, Supple. Fig.1A, OS: HR 1.39, 95% CI 0.98-1.97, P=0.065, Supple. Fig. 1B). Analyses using PS-adjusted data also showed no significant difference in RFS and OS between the type O and non-type O groups (Supplementary Fig. 1C). The 5-year RFS rate was 28.3% for type O and 24.9% for non-type O (HR: 1.15, 95% CI: 0.83-1.61, P = 0.405, Supple. Fig. 1C), and the 5-year OS rate was 60.3% for type O type and 51.8% for non-type O (HR: 1.38, 95% CI 0.88-2.16, P = 0.158, Supple. Fig. 1D).

Reviewer 2 Report
Thank you for the opportunity to review the prognostic impact of ABO blood group in hepatocellular carcinoma following hepatectomy by Kaibori et al. The study focused on ABO blood group-based prognosis in a population of patients undergoing surgical resection for hepatocellular carcinoma. In propensity score-matched A-type to non-A-type, the authors found a shorter recurrence-free survival profile and overall survival profile in A-type patients. The data analysis is partially flawed in that HCC biomarkers with clearly defined roles in biological aggressiveness are used at surveillance level binary thresholds. Control for continuous biomarker levels between the propensity score-matched datasets must be addressed.
Only propensity matching variable associated with HCC biology was AFP. AFP appears to have been propensity score matched based on an AFP < 10 ng/mL ≥. This is remarkably close to the upper limit of normal for AFP (8 ng/mL). The continuous AFP level above normal provides an excellent biomarker for biological aggressiveness and all the critical data is being lost with this threshold. The range of AFP levels cannot be discerned from the dataset and the question of adequate control of aggressive tumor biology at presurgical baseline remains. It is plausible that the A-type group, which had a higher percentage of multifocal disease, may also have significantly higher AFP levels, and thus anticipated poorer RFS outcomes following resection.
Figure 1 – what is the log-rank P-value in RFS and OS for the overall test of significance?
Table 3 – where the continuous levels for AFP and PIVKA-II associated with overall recurrence? If so, it would be interesting to see the receiver operating curves and optimal threshold values for AFP and PIVKA-II. Based on the size of the dataset, I would hypothesize these values to approach levels more consistent with those used to define aggressive biology in the literature as opposed to the surveillance threshold values in the manuscript analysis. This is critical, as mentioned above, because this information is being lost in the intergroup comparison and could be the driving force behind the blood-type restricted differences observed in the current analytical approach.
Author Response
Responses to the comments of Reviewer #2
Thank you for your valuable comments.
Major comments
- The data analysis is partially flawed in that HCC biomarkers with clearly defined roles in biological aggressiveness are used at surveillance level binary thresholds. Control for continuous biomarker levels between the propensity score-matched datasets must be addressed. Only propensity matching variable associated with HCC biology was AFP. AFP appears to have been propensity score matched based on an AFP < 10 ng/mL ≥. This is remarkably close to the upper limit of normal for AFP (8 ng/mL). The continuous AFP level above normal provides an excellent biomarker for biological aggressiveness and all the critical data is being lost with this threshold. The range of AFP levels cannot be discerned from the dataset and the question of adequate control of aggressive tumor biology at presurgical baseline remains. It is plausible that the A-type group, which had a higher percentage of multifocal disease, may also have significantly higher AFP levels, and thus anticipated poorer RFS outcomes following resection.
Response
Thank you for raising this point. Our study investigated the prognostic impact of ABO blood group in a population of patients with HCC who had undergone surgical resection. When calculating propensity scores for type A and non-type A patients, we assumed that the continuous biomarker levels had a skewed rather than normal distribution. Therefore, as a robust method, we divided the data into two categories according to the median and estimated the propensity score using a logistic regression model in which 'high' corresponded to biomarker levels equal to or above the median and 'low' if the levels were below the median.
As you note, when binarizing continuous data, an appropriate threshold value must be chosen and potential bias between datasets after matching should be considered. Since there was no significant difference between the two groups in terms of AFP and PIVKA-II levels, we believe that appropriate matching was performed.
- Figure 1 – what is the log-rank P-value in RFS and OS for the overall test of significance?
Response
Thank you for noting this omission. In the updated version of Figure 1 we added the log-rank P-values for RFS and OS determined for the overall test.
- Table 3 – where the continuous levels for AFP and PIVKA-II associated with overall recurrence? If so, it would be interesting to see the receiver operating curves and optimal threshold values for AFP and PIVKA-II. Based on the size of the dataset, I would hypothesize these values to approach levels more consistent with those used to define aggressive biology in the literature as opposed to the surveillance threshold values in the manuscript analysis. This is critical, as mentioned above, because this information is being lost in the intergroup comparison and could be the driving force behind the blood-type restricted differences observed in the current analytical approach.
Response
Thank you for raising this point. Many studies have already shown that AFP and PIVKA-II levels are associated with RFS and OS after hepatectomy for patients with HCC. The ROC analysis of OS for our study showed AFP and PIVKA-II thresholds of 15.1 and 1711, respectively. For AUC (95% CI), the value for AFP was 0.640 (0.88-0.691) and for PIVKA-II was 0.642 (0.590-0.694). Subject enrollment in this study was limited to patients with HCC who had undergone liver resection at specialized hospitals, which tend to have higher thresholds than those commonly used for screening and early diagnosis. Therefore, as a sensitivity analysis, we performed a Cox proportional hazards analysis using 8 ng AFP and 40 mAU/mL PIVKA-II as common thresholds instead of the ROC thresholds. The results showed a similar trend in threshold change for non-type A (RFS: HR 0.72, 95% CI 0.53-0.97, P = 0.029, OS: HR 0.57, 95% CI 0.38-0.84, P = 0.005), suggesting that there is an association between RFS and OS prognosis.
|
Variable |
Recurrence-free survival |
|
Overall survival |
|||||
|
HR |
(95% CI) |
P |
|
HR |
(95% CI) |
P |
||
|
Age (≥ vs. <73 years) |
1.04 |
(0.76–1.42) |
0.810 |
1.13 |
(0.76–1.68) |
0.554 |
|
|
|
BMI (≥ vs. <23 kg/m2) |
1.07 |
(0.78–1.46) |
0.685 |
1.08 |
(0.71–1.64) |
0.711 |
|
|
|
ASA-PS (Class III vs. II) |
1.32 |
(0.88–1.98) |
0.177 |
1.62 |
(1.00–2.63) |
0.051 |
|
|
|
HBsAg (Positive vs. Negative) |
0.71 |
(0.44–1.16) |
0.174 |
0.41 |
(0.19–0.86) |
0.018 |
|
|
|
HCV Ab (Positive vs. Negative) |
1.01 |
(0.72–1.41) |
0.949 |
0.81 |
(0.52–1.26) |
0.347 |
|
|
|
Platelet (≥ vs. <15.5 ×104/μL) |
1.12 |
(0.81–1.55) |
0.497 |
1.36 |
(0.88–2.11) |
0.164 |
|
|
|
ALT (≥ vs. <29 IU/L) |
0.92 |
(0.68–1.24) |
0.585 |
1.03 |
(0.70–1.53) |
0.884 |
|
|
|
Prothrombin time (≥ vs. <87 %) |
0.78 |
(0.55–1.10) |
0.158 |
0.78 |
(0.49–1.23) |
0.282 |
|
|
|
AFP (≥ vs. <8 ng/ml) |
1.21 |
(0.87–1.70) |
0.259 |
1.73 |
(1.06–2.82) |
0.027 |
|
|
|
PIVKA-II (≥ vs. <40 mAU/mL) |
1.70 |
(1.19–2.42) |
0.003 |
2.31 |
(1.37–3.87) |
0.002 |
|
|
|
ICGR15 (≥ vs. <14.8 %) |
0.97 |
(0.70–1.37) |
0.880 |
1.09 |
(0.70–1.70) |
0.711 |
|
|
|
ALBI score (Grade 2 vs. 1) |
1.43 |
(1.02–2.01) |
0.037 |
1.55 |
(0.98–2.46) |
0.062 |
|
|
|
ALBI score (Grade 3 vs. 1) |
0.73 |
(0.20–2.69) |
0.633 |
1.05 |
(0.26–4.29) |
0.948 |
|
|
|
Child-Pugh score (≥ vs. <6) |
1.38 |
(0.79–2.39) |
0.254 |
1.61 |
(0.90–2.87) |
0.111 |
|
|
|
Operative blood loss (≥ vs. <617 ml) |
1.21 |
(0.89–1.66) |
0.223 |
1.22 |
(0.81–1.85) |
0.346 |
|
|
|
Fibrosis stage (f0 or 1 or 2 or 3 vs. f4) |
0.94 |
(0.65–1.37) |
0.750 |
0.74 |
(0.45–1.22) |
0.243 |
|
|
|
Tumor size (≥ vs. <3.5 cm) |
1.09 |
(0.78–1.52) |
0.625 |
0.91 |
(0.59–1.40) |
0.672 |
|
|
|
Number of tumors (Multiple vs. Solitary) |
1.58 |
(1.14–2.20) |
0.006 |
1.28 |
(0.84–1.96) |
0.250 |
|
|
|
Portal vein invasion (Positive vs. Negative) |
1.57 |
(1.12–2.20) |
0.009 |
1.92 |
(1.19–3.09) |
0.008 |
|
|
|
Postoperative complications (Yes vs No) |
1.32 |
(0.90–1.93) |
0.154 |
2.26 |
(1.40–3.64) |
<0.001 |
|
|
|
Blood type (Non-A vs. A) |
0.72 |
(0.53–0.97) |
0.029 |
0.57 |
(0.38–0.84) |
0.005 |
|
|

Round 2
Reviewer 1 Report
The manuscript has been revised adequately.
Reviewer 2 Report
The authors have adequately responded to my comments.